# NF-κB Dependent Chemokine Signaling in Pancreatic Cancer

**DOI:** 10.3390/cancers11101445

**Published:** 2019-09-26

**Authors:** Claudia Geismann, Heiner Schäfer, Jan-Paul Gundlach, Charlotte Hauser, Jan-Hendrik Egberts, Günter Schneider, Alexander Arlt

**Affiliations:** 1Laboratory of Molecular Gastroenterology & Hepatology, Department of Internal Medicine I, UKSH-Campus Kiel, 24105 Kiel, Germany; cgeismann@email.uni-kiel.de (C.G.); hschaef@1med.uni-kiel.de (H.S.); 2Institute of Experimental Cancer Research, UKSH Campus Kiel, 24105 Kiel, Germany; 3Department of Surgery, UKSH-Campus Kiel, 24105 Kiel, Germany; jan-paul.gundlach@uksh.de (J.-P.G.); charlotte.hauser@uksh.de (C.H.); jan-hendrik.egberts@uksh.de (J.-H.E.); 4Technische Universität München, Klinikum rechts der Isar, II. Medizinische Klinik, 81675 Munich, Germany; guenter.schneider@tum.de

**Keywords:** NF-κB, chemokine, cytokine, pancreatic cancer, apoptosis, chemotherapy, microenvironment

## Abstract

Pancreatic cancer is one of the carcinomas with the worst prognoses, as shown by its five-year survival rate of 9%. Although there have been new therapeutic innovations, the effectiveness of these therapies is still limited, resulting in pancreatic ductal adenocarcinoma (PDAC) becoming the second leading cause of cancer-related death in 2020 in the US. In addition to tumor cell intrinsic resistance mechanisms, this disease exhibits a complex stroma consisting of fibroblasts, immune cells, neuronal and vascular cells, along with extracellular matrix, all conferring therapeutic resistance by several mechanisms. The NF-κB pathway is involved in both the tumor cell-intrinsic and microenvironment-mediated therapeutic resistance by regulating the transcription of a plethora of target genes. These genes are involved in nearly all scenarios described as the hallmarks of cancer. In addition to classical regulators of apoptosis, NF-κB regulates the expression of chemokines and their receptors, both in the tumor cells and in cells of the microenvironment. These chemokines mediate autocrine and paracrine loops among tumor cells but also cross-signaling between tumor cells and the stroma. In this review, we will focus on NF-κB-mediated chemokine signaling, with an emphasis on therapy resistance in pancreatic cancer.

## 1. Introduction

Pancreatic ductal adenocarcinoma (PDAC) represents the 4th leading cause of cancer-related deaths in western countries. The majority of PDAC arise from precursor lesions, pancreatic intraepithelial neoplasia (PanIN-1 to -3). In addition, PDAC can originate from mucinous cystic neoplasia (MCN), intraductal papillary mucinous neoplasia (IPMN), and from a process called acinar-ductal metaplasia (ADM), as recently shown in genetic mouse models [1]. Though a better understanding of the mechanisms of pancreatic carcinogenesis has been achieved over the years, therapeutic options in pancreatic cancer are still highly limited, and do not significantly improve life expectancy. One of the major problems of PDAC is a very difficult and late diagnosis. Consequently, only 15% of the patients have a localized disease status. For these patients, the five year survival prognosis after potentially curative surgical resection can be significantly increased to up to 50% through adjuvant chemotherapy [2]. The remaining 85% of the patients present with a locally advanced or metastatic tumor status. For patients with a locally advanced status, promising new neoadjuvante trials are under way [3], but for the majority of PDAC patients, only palliative radio- and/or chemotherapy options exists. After decades of treatments with 5-fluoruracil or gemcitabine, new therapeutic regimens based on combinations like FOLFIRINOX have recently been established, increasing the life expectancy of 6–7 months seen with gemcitabine therapy to 10–11 months. However, despite these recent advances, the overall outcome of patients remains desperately poor, and deciphering the mechanisms limiting the treatment options of PDAC is of pivotal importance. Among other factors, the two main determinants of this profound therapy resistance are: (I) the dysregulated activity of inflammation associated transcription factors, including NF-κB, Nrf2 and NFAT; and (II) a desmoplastic inflammatory microenvironment [4,5,6,7,8,9].

This microenvironment is one of the hallmarks of PDAC and distinguishes the tumor entity from many other cancers. Cellular compounds of this microenvironment, predominantly cancer-associated fibroblasts (CAFs), inflammatory and immune cells, but also non-cellular components like small blood vessels and extracellular matrix (ECM), profoundly compromise the desmoplasia that surrounds the malignant cells and occupies up to 80% of the entire tumor.

While numerous studies have documented a tumor-promoting function of the stromal key components (fibroblasts, endothelial cells, inflammatory cells and ECM proteins), and their therapeutic targeting was considered to be a promising anti-tumor strategy, recent data showed that these concepts have to be revisited carefully, particularly addressing the impact of each particular component [5,10]. As already mentioned, one additional major component of the microenvironment is inflammatory cells. These macrophages, T-cells, mast cells, and neutrophil granulocytes have been reported in opposing roles in the progression of PDAC. Using more complex genetic and organoid-derived models, we have to decipher this double-edged sword of immune cells in the biology of cancer progression and therapy resistance. Some inflammatory cells, particularly cytotoxic CD8^+^ T-cells, which are typically excluded from the tumor by a molecular shield, have the power to eliminate PDAC tumor cells, acting against tumor development and progression [11]. Other immune cells, particularly differentiated macrophages, contribute to tumor progression by paracrine loops. Since the majority of inflammatory cells in solid tumors are constituted by macrophages, derived either from recruited blood monocytes or resident tissue macrophages [9], such a protumorigenic activity of the inflammatory cells predominates. The composition and differentiation of the microenvironment is controlled by several inflammatory signaling/transcription pathways and chemokines. One of the best-established proinflammatory signaling pathways in pancreatitis and PDAC is the NF-κB pathway. An elevated basal and or inducible NF-κB activation is linked to many aspects of therapy-resistance, tumor cell proliferation and metastatic phenotype [8,12]. Besides direct regulation of the apoptotic cascade by some of the NF-κB target genes in PDAC [8,12,13,14,15,16,17], NF-κB controls a plethora of chemo- and cytokines, which orchestrate the complex interactions between all components of the PDAC microenvironment. This review will focus on published evidence of functional relevant NF-κB/chemokine interactions in PDAC with emphasis on therapeutic intervention options and the consequences of immunomodulatory therapies in PDAC.

## 2. NF-κB Signaling in PDAC

As outlined above, inflammation and tumorigenesis are often associated with the activation of the NF-κB signaling pathway. The number of NF-κB target genes is enormous and still growing, and the components and signaling molecules of these pathways were superbly reviewed in the NF-κB special issue of this journal [14,18,19,20]; that is why we will focus on some of the specific aspects in PDAC. NF-κB activity induces the secretion of cytokines and growth factors, contributing to tumor progression on one hand through the recruitment of immune cells, and on the other hand by cell intrinsic effects directly mediating apoptosis inhibition [21]. Several anti-apoptotic genes like the Bcl and IAP family members have been described as NF-κB target genes, directly blocking the apoptotic cascade at several points, conferring therapy-resistance [8]. Numerous reports have shown that death receptor ligands in general, but also chemotherapeutic drugs like etoposide and gemcitabine, activate NF-κB and directly attenuate their own apoptotic potential [16,17,22,23]; however, there is some controversy as to whether this inducible or constitutive NF-κB activity is more important for PDAC resistance. In addition to this direct effect of anti-apoptotic NF-κB target genes on PDAC cell survival, some apoptotic stimuli like TNF are also NF-κB target genes that provide auto- and paracrine death receptor ligand loops either through secretion of the death ligands by tumor cells or by infiltrating immune cells which lead to enhanced proliferation and/or survival [23,24]. Even if some chemotherapeutic drugs are able to induce NF-κB in a same manner as death-receptor ligands, the published evidence that constitutive NF-κB activity plays a major role in the resistance of PDAC against chemotherapeutic drugs is overwhelming [8,12]. In conclusion, dysregulated NF-κB signaling is a hallmark of PDAC development and chemoresistance mediated by cell intrinsic pathways (mainly anti-apoptotic genes) as well paracrine loop recruiting and influencing the microenvironment [8,9,12,21].

In this review, we will focus on the role of NF-κB controlled auto- and paracrine chemokine signaling in PDAC.

## 3. Chemokines and Their Receptors

Chemokines are a group of cytokines that mediate chemotaxis of their target cells. These small (approximately 8–17 kD) proteins bind to G-protein-coupled receptors and induce chemotaxis towards the highest chemokine concentration. Since chemokine receptors are mainly expressed on immune cells, these small molecules are major players in the control of immune cell migration. Recently, the expression of chemokine receptors has also been detected on epithelial tumor cells, leading to the conclusion that, next to the chemokine-mediated composition of the immune compartment of solid tumors, chemokines also regulate the migration and metastasis of the tumor cells (see below).

Approximately 50 Chemokines, which can be allocated into four subfamilies (CCL, CXCL, CX3CL, and XC, and their corresponding 19 receptors, are described (Table 1).

The number and position of the cysteine residues defines the systematic nomenclature: (1) The CC chemokine family has the first two cysteine residues adjacent to each other, (2) the CXC family has a single amino acid residue in between the first two cysteines (3) in the CX3C chemokine family three amino acid residues separate the first two cysteines, and (4) in the XC family there is only one cysteine residue and only one disulphide chain. As shown in Table 1, many chemokines—especially the CC chemokines—bind multiple receptors, and most receptors bind multiple chemokines. Furthermore, there are alternative names for nearly every chemokine, which sometimes make it difficult to do a proper literature research. However, one of the most striking problems in deciphering the role of the chemokines in disease is the promiscuity in the interaction of the chemokines with the receptors (Table 1). In this review, we will focus on published evidence for the lead chemokine in the outcome of PDAC cells/patients and chemokines clearly interacting with the NF-κB pathway (Table 2).

As already described, chemokines are acting as chemotactic modulators, recruiting immune cells and/or tumor cells along a gradient. However, there are effects of the chemokines independent of chemotaxis especially in cases of tumor autocrine loops (see below).

In addition to their biochemical properties, chemokines can be classified as “inducible/inflammatory” and “constitutive/homeostatic”. Most chemokines are inducible proteins that are part of inflammatory processes and function as danger signals to recruit and activate immune cells. The homeostatic chemokines are constitutively produced and regulate the composition and function of lymphoid organs and function as guardians of healthy tissue. Part of this group are CCL18, CCL19, CCL21, CXCL12, CXCL13 und CXCL14. Some of the chemokines (CCL1, CCL17, CCL20, CCL22, CCL25, CXCL9, CXCL10, CXCL11 und CXCL16) cannot be clearly allocated to one of these two groups.

In addition to these characteristics, the chemokine family receptors are thought to be predominantly expressed on specific target cells, explaining some of their biological functions. For instance, CC family receptors are mainly expressed on monocytes and lymphocytes, as well as eosinophilic and basophilic granulocytes. In contrast, CXC chemokines have a major impact on angiogenesis, and their receptors are predominantly expressed on neutrophils. The only known member of the CX3C family, Fraktalkine, has a major impact on the migration of T-lymphocytes and monocytes through interaction with endothelial cells. As discussed in the next paragraph, some of the effects of chemokines in PDAC can be attributed to these well-known features, but there are other cell-intrinsic as well as microenvironment-regulating mechanisms.

## 4. NF-κB Interacting Chemokines in PDAC

As outlined above, a plethora of chemokines are described in different aspects of PDAC carcinogenesis and therapy resistance. Since cancer in general induces a stress response, proinflammatory mediators are frequently upregulated, yet the effects on the tumor are still not well understood. In this review, we will focus on published evidence for the functional relevance of the chemokines in PDAC carcinogenesis (Table 1) and an interaction of the chemokine with the NF-κB pathway (Table 2).

### 4.1. CCL2

The chemokine CCL2 or monocyte chemotactic protein 1 (MCP1) is a proinflammatory chemokine that binds to CCR2 and CCR4. Early publications indicated that PDAC cells produce this chemokine under normal conditions [27], and this basal expression is further increased when the cells are stimulated with IL-1, TNF-α or FAS ligand [26]. Furthermore, the regulation of CCL2 expression in PDAC cells are attributed to the NF-κB pathway [25,26,27]. The functional relevance was not further investigated, but the authors speculated that CCL2 might recruit immune cells to PDAC tissue [26,27]. In contrast, a recent publication [25] raised the possibility that RRM2 induces invasiveness of PDAC through NF-κB and that CCL2 produced by regulatory T-cells in this context contribute to tumor proliferation and angiogenesis through the generation of an immunosuppressive microenvironment.

### 4.2. CCL5

CCL5 or RANTES binds to the CCR1, CCR3 and CCR5 receptors activating GPR75. These receptors are expressed on T-cells, monocytes and eosinophilic granulocytes. Furthermore, CCL5 is secreted by cytotoxic T-cells and part of the defense program against HIV by competing the binding of HIV to CCR5. Besides the observation that CCL5 is produced by PDAC cells in the same manner as CCL2 [27], a recent report elegantly showed that the secretion of RANTES and TNF-α through macrophages secrete RANTES/CCL5 and TNF-α in pancreatitis induced acinar-to-ductal metaplasia (ADM). RANTES/CCL5 in turn activates the NF-κB pathway and, through this, mediates survival, proliferation of PDAC cells and degradation of extracellular matrix [28].

### 4.3. CCL20

CCL20 or LARC interacts with its sole receptor CCR6. CCR6 is expressed on a variety of immune cells including macrophages, dendritic cells and T-cells as well as on different tumor cells [68,69,70]. The CCL20-CCR6 axis is one of the few samples of an exclusive chemokine/receptor interactions and has been implicated in cancer progression of several human carcinomas, including lung, colon, prostate, oral squamous cell carcinomas, leukemia, and melanoma [71,72]. For these tumor entities, using cell line and xenograft models, a role for CCL20 in cancer development has been established [73,74,75,76]. For PDAC, only limited data exist, showing an increased expression of CCL20 in pancreatic carcinoma tissues compared to the pancreatic control tissue significantly associated with advanced T-stage, [71,77,78]. Several reports have indicated that CCL20 has a direct impact on cancer cells [79,80,81,82,83]. We [32] and others [30] were able to show that CCL20 is a NF-κB target gene in PDAC cells. In contrast to the reported autocrine functions in colon carcinoma cells our data deciphered, we were able to decipher a RelA/CCL20-mediated onco-immuno-crosstalk, conferring resistance against death receptor ligand-induced apoptosis [29].

### 4.4. CCL21

CCL21/6Ckine elicits its effects by binding to CCR7 and fibroblastic reticular cells express this chemokine to attract naïve T-cells to the T-cell zone. In PDAC cell lines, CCL21 (and also CCL19) are described as target genes of the non-canonical NF-κB pathway [32]. Recently, a CCL21/CCR7 axis in CD133+ pancreatic cancer stem-like cells (CSC) has been reported [31]. In this context, CCR7 is strongly upregulated and binding of CCL21 induces metastasis by modulating EMT and the Erk/NF-κB pathways.

### 4.5. CXCL1

CXCL1/GRO1, GROα, KC is secreted by macrophages, neutrophils and tumor cells and mainly attracts neutrophils. It has been reported to be involved in angiogenesis, arteriogenesis, inflammation and wound healing, but also in many aspects of tumorigenesis. This chemokine elicits its effects by signaling through the chemokine receptor CXCR2. In PDAC, CXCL1 is described as being a part of the NDRG1/Cap43 regulated tumor suppressor pathway. NDRG1 inhibits IKK signaling in PDAC and thereby lowers the expression of the NF-κB target genes CXCL1, CXCL5 and IL-8/CXCL8. In summary, the authors suggest a novel mechanism by which NDRG1/Cap43 modulates tumor angiogenesis/growth and infiltration of macrophages/neutrophils through attenuation of chemokine expression. While this study suggests a tumor-promoting effect of CXCL1, another study describes a central role for this chemokine in oncogene-induced senescence (OIS). During carcinogenesis in the Kras driven PDAC-mouse model, RelA induces the expression of CXCL1 that mediates OIS through its interaction with CXCR2. Inactivation of the receptor leads to increased tumor proliferation and decreased survival of the mice. In summary, this work suggested a RelA/CXCL1/CXCR2 axis as a major guardian against PDAC tumor development [33].

### 4.6. CXCL5

CXCL5/ENA78 is induced by proinflammatory cytokines like IL-1β and TNF-α and can be inhibited by interferon gamma. Like other CXC family members, CXCL5 stimulates the chemotaxis of neutrophils possessing angiogenic functions. CXCL5 has been implicated in connective tissue remodeling and to regulate neutrophil homeostasis. The fact that CXCL5 uses the same receptor like CXCL1, CXCR2, offers an explanation for why parallel observations on these two chemokines in PDAC have been made. In line with this, the results described for the NDRG1-modulated IKK pathway in PDAC are the same for CXCL1 and CXCL5 [34,36]. However, Chao et al. described a specific role for CXCR2 ligands in a genetic mouse model of PDAC [35]. In this model, the authors were able to show that the CXCR2 ligands CXCL2 and CXCL5 are differentially expressed in tumor and stromal cells. While the CXCL2 expression is elevated in the stromal compartment, CXCL5 expression is associated with the mutant Kras status in tumor cells and regulated by NF-κB and in line upregulated in the tumor cells. Ablation of CXCR2 strongly reduced immune cell recruitment to the PDAC and led to a T-cell-dependent suppression of tumor growth. The authors conclude that a CXCL2/CXCL5/CXCR2 axis is important for an immunosuppressive microenvironment in PDAC and that targeting this axis could be a potential therapeutic approach [35].

### 4.7. CXCL8

CXCL8, or interleukin 8 (IL-8), is a chemokine that is produced by a plethora of cell types. Furthermore, IL-8 can bind to several receptors, including CXCR1 and CXCR2, with a higher affinity for the CXCR1 receptor. In addition to its angiogenic functions, IL-8 has two major functions—mediating phagocytosis and chemotaxis. IL-8 has a central role in the chemotaxis of a multitude of cell types, but especially the attraction of neutrophils. Besides neutrophils a wide range of other cell types also respond to by IL-8 makes this chemokine, making IL-8 to one of the key regulators of neutrophil driven inflammation. In addition to its two major functions—chemotaxis and phagocytosis, IL-8 exerts also profound angiogenic functions.

Early works showed that IL-8 is either constitutively overexpressed in PDAC or induced by several death receptor ligands, and that it contributes to the progression of PDAC [22,27,52,53,54]. The major factor accounting for this connection, both the high IL-8 expression constitutive as well as the inducible activity of IL-8 in PDAC, is primarily regulated by the RelA subunit of NF-κB—either activated constitutively or in response to a multitude of inducing stimuli [8,16,17,22,23,27,52].

Accordingly, natural compounds inhibiting NF-κB in preclinical or clinical studies strongly reduce IL-8 expression, demonstrating its relevance as one of the major NF-κB target genes in PDAC [43,44,45,46,47,48,50]. Despite this overwhelming evidence for a regulation of IL-8 through NF-κB in PDAC, only limited data on the functional relevance of this chemokine in PDAC exist.

The best characterized effect of IL-8 in PDAC is a direct proliferation induction on the tumor cells [37]. Furthermore, Song et al. [40] were able to show that gemcitabine induced CXCL8 counteracts the chemotherapeutic drug through inducing neovascularization in xenograft PDAC. Such a critical role for angiogenesis in PDAC was confirmed by another study [55]. The authors were able to show that IL-8 is mediating angiogenesis and bone marrow cell mobilization to increase PDAC tumor growth and that these functions depend on a NF-κB induced IL-8 expression.

### 4.8. CXCL10

CXCL10 or interferon gamma-induced protein 10 (IP-10) is secreted by several cell types in response to interferon gamma. Like other CXC family members, CXCL10 mediates angiogenesis and chemoattraction for a wide range of immune cells and angiogenesis. It binds to the CXCR3. The only known receptor is CXCR3.

For PDAC, only little research has been done on the NF-κB-controlled actions of CXCL10 in pancreatic cancer [56]. Kuhnemut et al. were able to show, in this context, that CXCL10 is part of a negative effect of CUX1 on NF-κB signaling in PDAC. The authors were able to decipher a complex mode of action of CUX1 on the RelA subunit, which leads to the downregulation of NF-κB-regulated chemokines, such as CXCL10, thereby antagonizing T-cell attraction and enhancing angiogenesis in vitro [56].

### 4.9. CXCL12

CXCL12, or stromal cell-derived factor 1 (SDF1), is, like many other of the CXC chemokines, ubiquitously expressed in many tissues and cell types and its signaling was observed in several cancers, including PDAC. CXCL12 is produced by alternative splicing in two variants, SDF-1α/CXCL12a and SDF-1β/CXCL12b. It acts strongly chemotactically, especially for lymphocytes and mesenchymal cells but it also regulates the expression of CD20 on B cells. Like several other members of the CXC family, CXCL12 plays an important role in angiogenesis by recruiting endothelial progenitor cells. CXCL12 binds to CXCR4 and CXCR7.

In the gastrointestinal tract, the CXCL12-CXCR4 axis is under investigation as an anti-fibrotic therapy [84] in the treatment of chronic pancreatitis and also in therapeutic interventions of several cancer entities.

As already discussed for CCL21, CXCL12 has been implicated in the non-canonical NF-κB signaling pathway in PDAC [32]. Interestingly, its receptor, CXCR4, has also been described as a NF-κB target gene in several types of epithelial cancers, including PDAC [63]. Moreover, in pancreatic cancer cells the expression of CXCL12 can be further upregulated by gemcitabine in a NF-κB-dependent fashion, and is subsequently associated with an increased proliferation and drug resistance [60,62]. The CXCR4 inhibitor AMD3100 can counteract this tumor promoting effect [62]. It is also notable that CXCL12 can also function as an inductor of the NF-κB signaling pathway in PDAC cell lines where stromal cell derived CXCL12 induces sonic hedgehog (SHH) upregulation [61]. In this preclinical model of PDAC, CXCL12 enhances the direct binding of NF-κB to the SHH promoter. This upregulation of SHH in the tumor cells in turn lead to higher CXCL12 expression in stromal cells, conferring a bidirectional tumor-stromal feedback loop for tumorigenesis [61]. Even if the authors did not investigate the regulation of CXCL12 in the stromal cells, a recent report clarified that in pancreatic stellate cells, CXCL12 expression is under control of NF-κB [59]. This observation was confirmed in knock-out cell lines for the p50 NF-κB subunit [58]. In this in vitro and in vivo model-based study, NF-κB/p50 activity regulates the expression of CXCL12 in pancreatic stellate cells, which in turn leads to an immunosuppressive microenvironment by the prevention of a cytotoxic T-cell infiltration into the tumor. Blocking CXCR4 with AMD3100 reversed this effect and increased antitumor immunity [58]. These complex interactions independent of direct effects on the epithelial cells highlight the importance of the CXCL12-CXCR4 axis in orchestrating the crosstalk between tumor cells and other components of the microenvironment like immune cells and fibroblasts, but also other components like the extracellular matrix. Understanding of this chemokine-mediated cross-talk is important for cancer immunotherapy [85], but also for understanding the influence of nodal status on microenvironment and prognosis [86].

### 4.10. CXCL14

CXCL14 is constitutively expressed at high levels in many normal tissues. Interestingly, fibroblast seems to be the major source of this chemokine and cancer cells exhibit a reduced or nearly absent expression. CXCL14 is chemotactic for monocytes, dendritic cells and NK cells. It is also inhibiting angiogenesis by blocking endothelial cell chemotaxis. In contrast to the distribution pattern in other normal and cancerous tissues, CXCL14 is upregulated in pancreatic cancer tissues compared to chronic pancreatitis and normal pancreas. However, direct stimulation of cancer cells with CXCL14 has no effect on cell viability or cell death. In contrast, the authors were able to show that CXCL14 increased invasiveness of pancreatic cancer cells by influencing the RelA subunit without affecting MMP-2 and VEGF secretion [64]. It is tempting to speculate that CXCL14 could be involved in the effects of pancreatic stellate cell/fibroblast-mediated resistance mechanism [87,88]. Such a function of fibroblast targeting CXCL14 loops has been reported for other cancers with profound microenvironment like breast cancer [89], but data on PDAC on this topic are missing.

### 4.11. CXCL16

CXCL16 is comparable to other chemokines of larger size (27.6 kDa), and harbors a cytoplasmic tail, containing a potential phosphorylation site and a transmembrane domain allowing its expression either as a cell surface-bound or as a soluble molecule. Its expression is induced by inflammatory cytokines like IFN-γ and TNF-α and induces the migration of T-cells and NK cells.

In PDAC, it has been reported to be involved in the Kras-mediated somatostatin receptor subtype 2 (SST2) down-regulation during tumorigenesis. In KRAS(G12D); sst2+/- mice, the heterogeneous sst2 loss leads to the activation of PI3K/AKT signaling. This in turn results in upregulated NF-κB activity and NF-κB dependent initiation and progression of neoplastic lesions. Interestingly, CXCL16 is involved in the activation of PI3K signaling to AKT and NF-kappaB,-κB and neutralization of the chemokine blocks carcinogenesis. In line with this, the chemokine and its receptor CXCR6 were significantly more highly expressed in PDAC tissues and surrounding acini than in healthy tissues, leading to the hypothesis that CXCL16 might be a therapeutic target for PDAC [65].

### 4.12. CX3CL1

CX3CL1, or Fractaline, contains multiple domains, is the only known member of the CX3 chemokine family, and interacts with the CX3CR1 receptor. Soluble CX3CL1 leads potently to chemotaxis of T-cells and monocytes, while the cell-bound chemokine promotes strong adhesion of leukocytes to activated endothelial cells.

In human PDAC specimen, CX3CL1 is highly expressed, along with AKT/p-AKT, BCL-xl and BCL-2. In this publication, CX3CL1 directly promoted proliferation and therapy resistance of pancreatic cancer cells by leading to enrichment of pancreatic cancer cells in S phase with a concomitant decrease of the number of cells in G1 phase [67]. In contrast, we were not able to show a tumor cell intrinsic function of the chemokine but confirmed through a genome-wide unbiased approach that NF-κB/RelA regulates the expression of CX3CL1 in PDAC tumor cells. As already described for CCL20 [29], NF-κB-driven CX3CL1 expression had no direct effects on cancer cell apoptosis or proliferation. In contrast, the chemokine acts in a paracrine fashion, leading to an increased recruitment of inflammatory cells, which in turn reduces death receptor-induced apoptosis of PDAC cell lines [66].

## 5. Discussion

The link between NF-κB signaling and many aspects of cancer has been well established for more than two decades, since it was first noted that NF-κB is capable of inhibiting TNF-α-induced apoptosis. However, it took several years to establish chronic inflammation as one of the hallmarks of cancer [90], although inflammation is now recognized as one of the major causes of cancer. Since NF-κB is one of the central transcriptional regulators of many aspects of carcinogenesis, it is not surprising that this signaling pathway gained particular attention with regard to the concept of targeting inflammation driven carcinogenesis [91]. Among the enormous number of NF-κB target genes that could be involved in inflammation driven carcinogenesis, chemokines and chemokine receptors are of the greatest interest, since neutralizing and or receptor-blocking antibodies could easily be introduced into clinical trials. In addition to direct effects on tumor cells that make use of autocrine chemokine loops for proliferation and survival (see above), chemokines mediate immune cell trafficking into the tumors and hereby influence the composition of the tumor microenvironment. Therefore, direct and indirect effects of chemokine–chemokine–receptor interactions on the tumor cells may influence the outcome of the disease under conventional as well as classical but also under innovative cancer-immunotherapy. One of the problems in targeting these interactions is the high promiscuity of some of the chemokine-receptors-chemokines interactions (Figure 1). The specific blockage of a receptor might lead to the binding of the elevated chemokine to a lower affinity receptor with unpredictable outcomes for the patients [85].

PDAC represents the malignancy with the worst prognosis, and a profound desmoplastic tumor microenvironment considerably adds to this desperate condition. Originally, fibroblasts, called pancreatic stellate cells, were considered to be the major source of the protumorigenic and therapy-limiting functions of the tumor cell surroundings [10,92]. However, with the rising status of cancer immunotherapy since 2013, more and more attention has been paid to the immune compartment, also in PDAC research [93]. This immune compartment is thought to be highly immunosuppressive, and its composition is dominated by regulatory T-cells (Tregs), tumor-associated macrophages (TAMs), and myeloid-derived suppressive cells (MDSCs). These cells are able to block CD8 T-cell-mediated tumor recognition and clearance thereby contributing to the profound resistance against novel therapies based on new therapeutic interventions like checkpoint inhibitor targeting or inhibitors and cancer vaccines [93]. Therefore, understanding the role of signaling pathways that regulate the composition and/or function of the immune compartment of PDAC is pivotal to increase the efficacies of novel but also of already established therapeutic interventions.

### Anti-Tumor Effects of NF-κB/Chemokine Interactions—A Double Edged Sword

Interestingly, only very limited data on NF-κB/chemokine interactions leading to an immunosuppressive microenvironment exist [25,35,36]. In particular, the CXCR2-ligands CXCL5 and CXCL2 contribute to neutrophil accumulation and inhibit T-cell-dependent suppression of tumor growth [35]. In light of the corresponding literature, it is tempting to speculate that blocking CXCR2 could be a potential way to treat PDAC, but as already outlined above, the high number of different ligands on the chemokine receptors make these therapy strategies unpredictable in their outcome. In fact, for the CXCR2 ligand a central role for oncogene induced senescence of the CXCR2 ligand recently was reported recently [33].

In the Kras mouse model, pancreas-specific inactivation of CXCR2 prevented OIS, correlating with increased tumor proliferation and decreased survival thus challenging the role of blocking CXCR2 to increase anti-tumor therapy. Hence, inhibition of the CXCL2-CXCR2 axis is a double-edged sword and should be followed with high caution (Figure 2). Finally, IL-8/CXCL8 also binds to CXCR2 but also to numerous other none chemokine receptors. Given the considerable promiscuity of many receptor-chemokine systems, any prediction of the outcome of inhibiting a specific receptor or chemokine is quite difficult. Moreover, even for chemokines and receptors with one ligand-one receptor interactions, the exact role of the chemokine in PDAC can only be envisioned with difficulty, as highlighted by the data on CCL20 (Figure 3) [29,68,78,94,95].

CCL20 binds exclusively to CCR6 and, up to now, no other ligand for CCR6 exists. Nevertheless, there are reports that CCL20 is responsible for an autocrine loop leading to increased proliferation and survival of PDAC cells independent of immune-cell interaction [68]. Furthermore, several reports indicated that Th17 and dendritic cells recruited by CCL20-CCR6 interactions could promote tumor immunity and may lead to better tumor cell clearance [85]. However, recent reports [29,95] showed that CCL20-CCR6 is part of a NF-κB/chemokine onco-immuno crosstalk that leads to CCR6 dependent immune cell recruitment, which in turn promotes tumor cell invasion, proliferation and resistance against cell death inducing treatments.

These data highlight the importance of using more precise sophisticated models to investigate the function of NF-κB/chemokine signaling in PDAC, since cell culture models and/ or non-immune competent animal models too greatly simplify them and cannot reflect the complex interactions between tumor cells and tumor microenvironment. This problem is evident when looking at the excessive data of direct effects of the chemokines on PDAC cells discussed in this review. Many of these simplified findings will be certainly challenged when using immune competent mice, organoids and or sophisticated 2D- or 3D-coculture models, but this way of scrutinization is greatly needed for introducing chemokine targeting as therapeutic option in PDAC.

## 6. Conclusions

Though the search for more specific markers and diagnostic measures to detect PDAC earlier is the most relevant issue in overcoming the desperate conditions for the affected patients, it cannot be expected that a major breakthrough will occur in the near future. Therefore, a more sophisticated approach for treating patients with advanced PDAC remains the only option so far to improve their survival. Recent developments dealt with novel combinations of conventional anti-cancer drugs together with biologicals or small compounds specifically targeting certain cancer cell pathways (TKs, EGF, ras/raf, PI3K-Akt, etc.). However, the benefit for PDAC patients remained overall quite modest. This also includes strategies interfering with the NF-κB pathway. Keeping in mind that this pathway has a pivotal role in many aspects in tumorigenesis it is obvious that only the full appreciation of all these actions would allow to assess the versatility of NF-κB as target in PDAC therapies.

One important, yet not well understood, modality is the link between NF-κB and the chemokine-mediated/immune modulatory network in cancers, giving rise of the complex interaction between the great variety of immune and stromal cells and the tumor cells. Deciphering all the facets of these interactions will essentially contribute to the understanding of the protumorigenic role of NF-κB and how the cancer cell is interconnected with the microenvironment through NF-κB regulated chemokines. Especially the poorly investigated role of NF-κB/chemokine regulated composition of the microenvironment seems to be more relevant than autocrine chemokine loops in PDAC. Based on these new insights then, more predictable therapy strategies could be delineated by targeting those parts of the interaction being more exclusive and lacking redundant or alternative pathways. As outlined above, extensive work is still needed to fully clarify this issue.

## Figures and Tables

**Figure 1 cancers-11-01445-f001:**
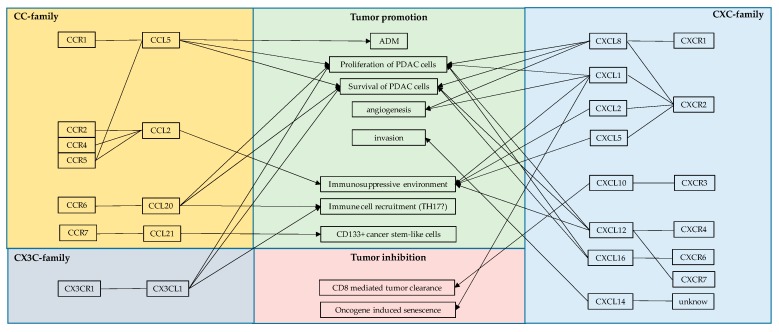
Schematic representation of the NF-κB/chemokine interactions in PDAC. Illustration of the interaction of the chemokines with their receptors (left and right) and the resulting effect on the tumor (middle).

**Figure 2 cancers-11-01445-f002:**
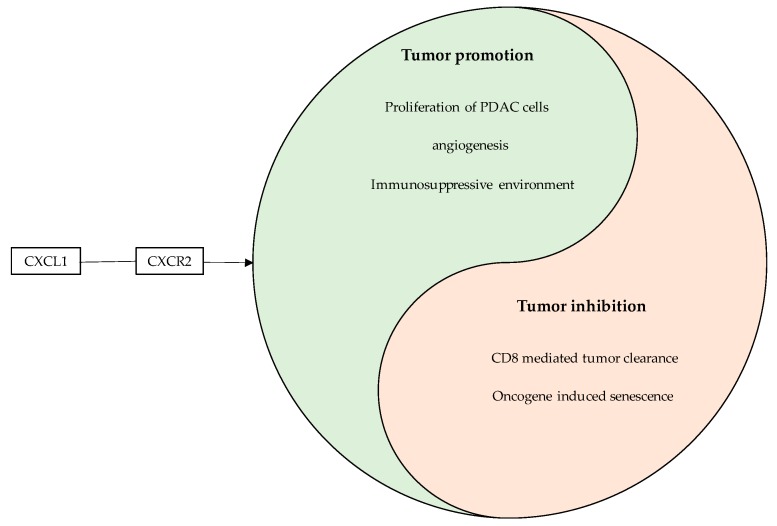
Contradictory effects of CXCL1-CXCR2 signaling in PDAC. Illustration of the Yin and Yang like effects of the CXCL1/CXCR2 interactions.

**Figure 3 cancers-11-01445-f003:**
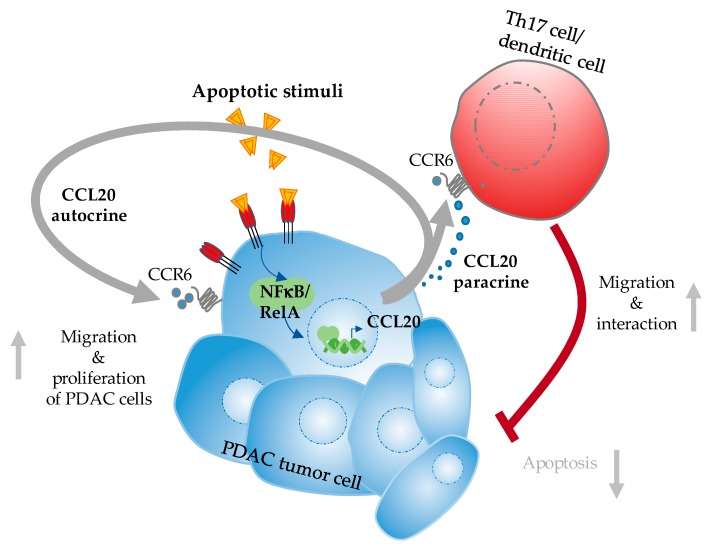
CCL20/CCR6 auto-and paracrine interactions in PDAC. Illustration of the interactions of the tumor cells with immune cells (right part of Figure 3) and the autocrine loop mediated by NF-κB controlled CCL20 release of the tumor cells.

**Table 1 cancers-11-01445-t001:** Chemokine families, their members, alternative names and interacting receptors are listed.

Family	Name	Alternative Name	Receptor
CC	CCL1	I-309, TCA-3	CCR8
	CCL2	MCP-1	CCR2/4/5
	CCL3	MIP-1α	CCR1/5
	CCL4	MIP-1ß	CCR5
	CCL5	RANTES	CCR1/3/5
	CCL6	C10, MRP-2	CCR1
	CCL7	MARC, MCP-3	CCR1/2
	CCL8	MCP-2	CCR8
	CCL9/10	MRP-2, CCF18	CCR1
	CCL11	Eotaxin	CCR3
	CCL12	MCP-5	CCR2
	CCL13	MCP-4, NCC-1, Ckß10	CCR1/2
	CCL14	HCC-1, MCIF, Ckß1, NCC-2, CCL	CCR1
	CCL15	Leukotactin-1, MIP-5, HCC-2, NCC-3	CCR1/3
	CCL16	LEC, NCC-4, LMC, Ckß12	CCR1/3
	CCL17	TARC, dendrokine, ABCD-2	CCR4
	CCL18	PARC, DC-CK1, AMAC-1, Ckß7, MIP-4	CCR8/GPR30
	CCL19	ELC, Exodus-3, Ckß11	CCR7
	CCL20	LARC, Exodus-1, Ckß4	CCR6
	CCL21	SLC, 6Ckine, Exodus-2, Ckß9, TCA-4	CCR7
	CCL22	MDC, DC/ß-CK	CCR4
	CCL23	MPIF-1, Ckß8, MIP-3, MPIF-1	unknown
	CCL24	Eotaxin-2, MPIF-2, Ckß6	CCR3
	CCL25	TECK, Ckß15	CCR9
	CCL26	Eotaxin-3, MIP-4α, IMAC, TSC-1	CCR3
	CCL27	CTACK, ILC, Eskine, PESKY, skinkine	CCR10
	CCL28	MEC	CCR10
CXC	CXCL1	Gro-α, GRO1, NAP-3	CXCR2
	CXCL2	Gro-ß, GRO2, MIP-2a	CXCR2
	CXCL3	Gro-γ GRO3, MIP-2ß	CXCR2
	CXCL4	PF-4	unknown
	CXCL5	ENA-78	CXCR2
	CXCL6	GCP-2	CXCR1/2
	CXCL7	NAP-2, CTAPIII, ß-Ta, PEP	CXCR2
	CXCL8	IL-8, NAP-1, MDNCF, GCP-1	CXCR1/2
	CXCL9	MIG, CRG-10	CXCR3
	CXCL10	IP-10, CRG-2	CXCR3
	CXCL11	I-TAC, ß-R1, IP-9	CXCR3
	CXCL12	SDF-1, PBSF	CXCR4/7
	CXCL13	BCA-1, BLC	CXCR5
	CXCL14	BRAK, bolekine	unknown
	CXCL15	Lungkine, WECHE	unknown
	CXCL16	SRPSOX	CXCR6
	CXCL17	DMC, VCC-1	unknown
C	XCL1	Lymphotactin α, SCM-1α, ATAC	XCR1
	XCL2	Lymphotactin ß, SCM-1ß	XCR1
CX3C	CX3CL1	Fractalkine, Neurotactin, ABCD-3	CX3CR1

**Table 2 cancers-11-01445-t002:** Chemokine/NF-κB pathway interactions in PDAC with the corresponding literature.

Family	Name	Alternative Name	Receptor	PDAC
CC	CCL2	MCP-1	CCR2/4/5	[25,26,27]
	CCL5	RANTES	CCR1/3/5	[27,28]
	CCL20	LARC, Exodus-1, Ckß4	CCR6	[29,30]
	CCL21	SLC, 6Ckine, Exodus-2, Ckß9, TCA-4	CCR7	[31,32]
CXC	CXCL1	Gro-α, GRO1, NAP-3	CXCR2	[30,32,33,34]
	CXCL2	Gro-ß, GRO2, MIP-2a	CXCR2	[35]
	CXCL5	ENA-78	CXCR2	[34,35,36]
	CXCL8	IL-8, NAP-1, MDNCF, GCP-1	CXCR1/2	[22,27,34,37,38,39,40,41,42,43,44,45,46,47,48,49,50,51,52,53,54,55]
	CXCL10	IP-10, CRG-2	CXCR3	[56]
	CXCL12	SDF-1, PBSF	CXCR4/7	[32,57,58,59,60,61,62,63]
	CXCL14	BRAK, bolekine	unknown	[64]
	CXCL16	SRPSOX	CXCR6	[65]
CX3C	CX3CL1	Fractalkine, Neurotactin, ABCD-3	CX3CR1	[66,67]

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
