# Peer review of "NF-κB Dependent Chemokine Signaling in Pancreatic Cancer"

_cancers, 2019, doi:10.3390/cancers11101445_

Round 1
Reviewer 1 Report
The manuscript entitled "NF-κB dependent chemokine signalling in pancreatic cancer" by Claudia Geismann, Heiner Schäfer, Jan-Paul Gundlach, Charlotte Hauser, Jan-Hendrik Egberts, Günter Schneider, and Alexander Arlt describes the relationships between chemokines and NF-κB in PDAC. The article is comprehensive and contains chemokine characteristics, which is well-arranged. The importance of chemokine-NF-κB interactions is also underlined when considering therapeutic approaches.
There are minor improvements needed:
1) Line 95: please specify either official gene or protein symbols. 2) Line 109: when defining IL-8, it would be better to define it here as a proinflammatory chemokine. 3) Line 286: please provide the literature. 4) Line 403 and Fig.3: DC as dendritic cell/s abbreviation does not need to be written as DC cell/s. 5) There are text modifications which should be deleted (lines: 214, 222, 238, 247, 252, 256, 288, 289, 294, 298, 307, 323, 325, 350). 6) Lines 89, 92: k-->κ. 7) Text text should be rechecked for the proper edition (lines: 198, 268, 270, 290).
Overall, this manuscript deserves to be published after introduction of minor changes.
Author Response
Point-by-point response
Both reviewers suggested only minor revisions which we addressed point by point below. In addition we revised the overlaps.
Reviewer 1
Line 95: please specify either official gene or protein symbols.We rewrote the paragraph and addressed this point
Line 109: when defining IL-8, it would be better to define it here as a proinflammatory chemokine.We rewrote the paragraph and addressed this point
Line 286: please provide the literature.We included the citation: Neesse A, Ellenrieder V (2017) NEMO-CXCL12/CXCR4 axis: a novel vantage point for antifibrotic therapies in chronic pancreatitis? Gut 66:211-212
Line 403 and Fig.3: DC as dendritic cell/s abbreviation does not need to be written as DC cell/s.We changed the paragraph and figure.
There are text modifications which should be deleted (lines: 214, 222, 238, 247, 252, 256, 288, 289, 294, 298, 307, 323, 325, 350).We are very sorry for these problems which was included in the editing of the internal review process. We completely checked the manuscript again and revised it.
Lines 89, 92: k-->κ. text should be rechecked for the proper edition (lines: 198, 268, 270, 290).See above

Reviewer 2 Report
Geismann et al. summarize NF-κB pathway role in PDAC, regarding the tumor cell intrinsic and the microenvironment mediated therapeutic resistance In this review, they focus on NF-κB mediated chemokine signalling and the related therapeutic window.
To the authors
The manuscript is well written, nonetheless, there are few sections that might deserve to be slightly restructured, in order to achieve the level and comprehensive overview that a journal like Cancers would aim to.
Minor points to consider in subsequent versions:
Page 3; in paragraph “Chemokines and their receptors”: when discussing the paper by the group of Wang (reference 55); I think it is important to mention that that particular study refers to CXCR4 and NF-kB; nonetheless, a tight correlation exists between immune-infiltrate and chemokines, angiogenesis and cancer progression and dissemination to distant sites and to nodal compartment. Because of these intimate interactions, the capacity of microenvironmental cells and dendritic cells can be also briefly discussed, since several examples have been recently published (i.e. PMID: 28555670; PMID: 31277479). These are important data and could expand the translational landscape of the above-mentioned data.
Finally, at Page 8; in paragraph "CXCL14": when discussing the paper by the group of Wente (reference 62) it would be worth to point out that the PDAC microenvironment and the cancer associated fibroblasts in particular strongly influence the chemokine mediated pro-tumor milieu: from a preclinical standpoint, several cancers with terribly poor prognosis could benefit from novel insights derived from this data (i.e.; PMID: 26284509; PMID: 30866547).
Author Response
Point-by-point response
Both reviewers suggested only minor revisions which we addressed point by point below. In addition we revised the overlaps.
Reviewer 2
Page 3; in paragraph “Chemokines and their receptors”: when discussing the paper by the group of Wang (reference 55); I think it is important to mention that that particular study refers to CXCR4 and NF-kB; nonetheless, a tight correlation exists between immune-infiltrate and chemokines, angiogenesis and cancer progression and dissemination to distant sites and to nodal compartment. Because of these intimate interactions, the capacity of microenvironmental cells and dendritic cells can be also briefly discussed, since several examples have been recently published (i.e. PMID: 28555670; PMID: 31277479). These are important data and could expand the translational landscape of the above-mentioned data.We completely agree that the tight correlation exists between immune-infiltrate and chemokines, angiogenesis and cancer progression and dissemination to distant sites and to nodal compartment is a rising topic in PDAC treatment and already discussed this point in the conclusions/discussions. However to highlight this aspect in this paragraph we included both mentioned manuscript and discussed them in the context.
Finally, at Page 8; in paragraph "CXCL14": when discussing the paper by the group of Wente (reference 62) it would be worth to point out that the PDAC microenvironment and the cancer associated fibroblasts in particular strongly influence the chemokine mediated pro-tumor milieu: from a preclinical standpoint, several cancers with terribly poor prognosis could benefit from novel insights derived from this data (i.e.; PMID: 26284509; PMID: 30866547).We included the mentioned publication and discussed the possible correlation – however published data are missing on CXCL14 in pancreatic fibroblasts and the aspect of this important cell type was discussed in another section of the review.

Reviewer 3 Report
Comments to the authors:
I am reviewing this review article following a first round of revisions. I want to congratulate the authors on a well done, comprehensive overview of NF-KB signalling and its role in PDAC.
I have two major comments
The target cells that the chemokine acts on in the progression of the disease- (Immune or the epithelial tumor cells )is a key factor. I would like the authors to add another column to Table 2 indicating the major target cell type bound by the respective chemokines. Outline a brief summary key for the figures (1-3) for improving clarity to the readers in addition to the figure titles.There are several missing propositions and inaccuracies in English which also needs to be corrected.